# Fecal Microbiota and Hair Glucocorticoid Concentration Show Associations with Growth during Early Life in a Pig Model

**DOI:** 10.3390/nu14214639

**Published:** 2022-11-03

**Authors:** Francesc González-Solé, David Solà-Oriol, Sandra Villagómez-Estrada, Diego Melo-Durán, Laura Victoria López, Nathaly Villarroel Román, Marina López-Arjona, José Francisco Pérez

**Affiliations:** 1Animal Nutrition and Welfare Service (SNIBA), Department of Animal and Food Science, Autonomous University of Barcelona, 08193 Bellaterra, Spain; 2Carrera de Medicina Veterinaria, Escuela Superior Politécnica de Chimborazo (ESPOCH), Riobamba 060155, Ecuador; 3Interdisciplinary Laboratory of Clinical Analysis of the University of Murcia (Interlab-UMU), Regional Campus of International Excellence ‘Campus Mare Nostrum’, University of Murcia, Campus de Espinardo s/n, 30100 Murcia, Spain

**Keywords:** lactation growth, nursery growth, pig, short-chain fatty acids, cortisol, cortisone, fecal microbiota

## Abstract

Identifying characteristics associated with fast or slow growth during early life in a pig model will help in the design of nutritional strategies or recommendations during infancy. The aim of this study was to identify if a differential growth during lactation and/or the nursery period may be associated with fecal microbiota composition and fermentation capacity, as well as to leave a print of glucocorticoid biomarkers in the hair. Seventy-five commercial male and female pigs showing extreme growth in the lactation and nursery periods were selected, creating four groups (First, lactation growth, d0–d21; second, nursery growth, d21–d62): Slow_Slow, Slow_Fast, Fast_Slow, and Fast_Fast. At d63 of life, hair and fecal samples were collected. Fast-growing pigs during nursery had higher cortisone concentrations in the hair (*p* < 0.05) and a tendency to have a lower cortisol-to-cortisone ratio (*p* = 0.061). Both lactation and nursery growth conditioned the fecal microbiota structure (*p* < 0.05). Additionally, fast-growing pigs during nursery had higher evenness (*p* < 0.05). Lactation growth influenced the relative abundance of eight bacterial genera, while nursery growth affected only two bacterial genera (*p* < 0.05). The fecal butyrate concentration was higher with fast growth in lactation and/or nursery (*p* < 0.05), suggesting it has an important role in growth, while total SCFA and acetate were related to lactation growth (*p* < 0.05). In conclusion, piglets’ growth during nursery and, especially, the lactation period was associated with changes in their microbiota composition and fermentation capacity, evidencing the critical role of early colonization on the establishment of the adult microbiota. Additionally, cortisol conversion to cortisone was increased in animals with fast growth, but further research is necessary to determine its implications.

## 1. Introduction

Early life experiences and environmental conditions have a critical role in the growth and development of mammals. On the one side, the early life microbial colonization of the gut, shaped by factors such as host genetics, environment, diet, immunological pressure, and antibiotics [1,2,3], may influence the programming of the mucosal immune response and the development of the gut barrier function, conditioning their propensity to develop certain health disorders [1,2,3]. Evidence suggests that a disruption of microbial colonization might cause lifelong deficits in growth and development [4]. On the other hand, early life stress has also been linked to physical and psychological sequelae later in life, including alterations in the immune system [5,6]. Extended exposure to glucocorticoids leads to increases in the sympathetic nervous system, hypothalamic pituitary adrenal axis, and inflammatory markers [5,6] and has been associated with an impairment of growth and development both in humans and animal models [7,8]. Therefore, studying the differences in the microbial colonization process and the experienced stress during early life among individuals with differential growth responses might help to understand which conditions are associated with optimal growth during this period.

Humans and pigs share a great physiological similarity in digestive and associated metabolic processes. In fact, the neonatal piglet has become a good model for the study of pediatric nutrition and metabolism [9]. In addition, it is well-established as a model for assessing interactions between microbiota and health, as it exhibits similar scenarios to humans, such as weaning diarrhea [9]. The individual growth response of piglets during their early life on a commercial farm is affected by multiple factors. During lactation, colostrum and milk intake are the primary drivers of piglet growth [10]. Initial differences in the birth BW and vitality among piglets will influence the sibling competition for the sow’s nutrients and may have a major effect on survival and growth [11,12]. Piglet’s weaning is a stressful event involving separation from the sow, an abrupt change in diet from sow’s milk to a plant-based diet, and the establishment of new social hierarchies in the nursery pens [13]. Afterward, the postweaning growth will highly depend on the management and nutritional strategies conducted to accelerate piglet adaptation to the solid diet and the new housing conditions [14,15]. When the stressful stimuli of weaning overpower a piglet’s adaptation, it can lead to chronic stress, poor performance, and increased mortality [14,16,17]. Piglets belonging to the same farrowing and weaning batch are exposed to very similar housing and environmental conditions, and they share a large part of their genetic background. These similarities and the challenges they face during the lactation and nursery periods make them a good model to study the factors that determine their individual growth responses and development during early life. Thus, identifying the characteristics of pigs with different growth rates during these periods is the first step to understanding those mechanisms and to designing individualized strategies to stimulate the development of slow-growing individuals.

In this study, it was hypothesized that gut microbiota, as well as the experienced stress during the lactation and nursery periods, might have a relationship with the piglets’ individual growth responses during these periods. Therefore, the aim of this study was to identify associations among the growth performance during the lactation and nursery periods with the fecal microbiota composition, its fermentation capacity, and the levels of glucocorticoid biomarkers in their hair.

## 2. Materials and Methods

### 2.1. Animals, Housing and Diet

The study was conducted in the farrowing room of a commercial farm from farrowing day to the day of weaning (d21) and in the nursery unit of the same farm during the 41 days post-weaning (d62).

A total of 324 commercial male and female piglets ((Landrace × Yorkshire) × Pietrain) from a single weaning batch (weaned at d21 ± SD 0.8 days of life) were weighed at birth (d0) and at d21 of lactation (weaning). A subset of 70 pigs with slow growth during lactation (139 ± SD 20.4 g/d, ranging from 97 g/d to 173 g/d, belonging to the 10–42% percentile of the initial population) and 81 pigs with fast growth during lactation (248 ± SD 21.2 g/d, ranging from 208 g/d to 295 g/d, belonging to the 60–93% percentile of the initial population) were weighed again at the end of nursery period (d62 of life). Then, pigs with slow growth during the nursery period from each subset were selected (6–30% percentile from the slow lactation subset and 5–26% percentile from the fast lactation subset) and also pigs with fast growth during the nursery period (76–100% percentile from the slow lactation subset and 79–100% percentile from the fast lactation subset), obtaining four groups as a factorial design: slow growth during the lactation and nursery periods (Slow_Slow; *n* = 18; 9 males + 9 females), slow growth during lactation but fast during the nursery period (Slow_Fast; *n* = 19; 10 males + 9 females), fast growth during lactation but slow during the nursery period (Fast_Slow; *n* = 19; 9 males + 10 females), and fast growth during the lactation and nursery periods (Fast_Fast; *n* = 19; 10 males + 9 females). The growth characteristics of each group are described in Table 1.

Sows and their litters were housed in individual farrowing pens (2.6 × 1.8 m^2^) on a partially slatted floor with a heated floor pad for piglets, equipped with a farrowing crate, an individual feeder, and nipple drinkers for sows and piglets. The temperature in the farrowing room was automatically controlled (22–24 °C). Water and feed were offered ad libitum to the sows. Piglets had access to water and were offered creep feed (2558 kcal NE/kg, 19.7% CP, and 1.370 digestible Lys) since d7 of lactation.

At weaning (d21), piglets were moved to the nursery unit without transport. Piglets were allocated in 2 pens blocked by sex (280 pigs/pen) and mixed with other animals that were not monitored since birth. Each pen (60 m^2^) was equipped with eight commercial feeders (Tolva EVO-800D, Porinox, Olot, Spain) and eight nipple bowl drinkers to provide ad libitum access to feed and water. The floor was completely slatted, and the temperature and ventilation rates were controlled using central and forced ventilation with an automatic cooling system. The first two days post-weaning, all animals were offered the creep feed diet and then were fed a pre-starter diet (2430 kcal NE/kg, 18.94% CP, and 1.29% digestible Lys) until d10 post-weaning. Afterward, a starter diet (2500 kcal NE/kg, 17.5% CP, and 1.18% digestible Lys) was offered until the end of the nursery period. All diets were formulated to meet the requirements for the maintenance and growth of newly weaned piglets [18] (Table 2).

### 2.2. Sample Collection

At the end of the nursery period (d63), fecal and hair samples were collected from the piglets included in the study. An aliquot of the feces was stored in 2 mL sterile cryotubes for analysis of the fecal microbiota, and another aliquot of 5 g was stored in Ziplock bags for the analysis of SCFA. Both aliquots were snap-frozen in dry ice and afterward kept at −80 °C. Hair samples were obtained with scissors from the lumbar area, cutting it as close to the skin as possible.

### 2.3. Cortisol and Cortisone Hair Concentration Analysis

The protocol used for cortisol and cortisone extraction was performed as described by López-Arjona et al. (2020) [19]. Briefly, the hair samples were weighed (250 mg), placed in a polypropylene tube, and covered with isopropanol (5 mL). The tube was mixed at room temperature (RT), centrifuged (1500× *g*, 1 min), and the isopropanol discarded. The samples were washed again with isopropanol and left at RT until completely dry. Next, the hair was pulverized to a fine powder in a homogenizer (Precellys Evolution homogenizer, Bertin Technologies, France) and incubated with 1 mL of methanol for 18 h at RT with continuous gentle agitation for steroid extraction. Samples were then centrifuged (2000× *g*, 5 min). The samples were evaporated to dryness in a Speed Vac Concentrator (Concentrator 5301, Eppendorf). The dry extracts were reconstituted with 0.1 mL of phosphate-buffered saline (PBS) and stored at –80 °C until analysis. Cortisol and cortisone hair concentrations were measured by sensitive assays based on AlphaLISA technology that also enabled the estimation of 11β-hydroxysteroid dehydrogenase type 2 activity [19]. The assay validation was performed as described by López-Arjona et al. (2020) [19]. The assays were precise (imprecision <12%) and accurate (recovery range, 80–115%) for cortisol and cortisone determination.

### 2.4. Short-Chain Fatty Acid Analysis

Short-chain fatty acids and lactic acid determinations were performed on feces by gas liquid chromatography. Sequentially, the samples were submitted to an acid–base treatment followed by an ether extraction and derivatization with N-(tertbutyldimethylsilyl)-N-methyl-trifluoroacetamide (MBTSTFA) plus 1% tertbutyldimethylchlorosilane (TBDMCS) agent, using the method of Richardson et al. (1989) [20], modified by Jensen et al. (1995) [21]. The volatile fatty acids measured were acetate, propionate, butyrate, iso-butyrate, valerate, iso-valerate, succinate, and formate.

### 2.5. 16S rRNA Gene Sequencing

The composition and structure of the microbial communities present in the fecal samples preserved at −80 °C were determined through a 16S rRNA gene sequence-based analysis. Bacterial DNA was extracted from 250 mg of each fecal sample using the commercial MagMAX CORE Nucleic Acid Purification Kit 500RXN (Thermo Fisher, Barcelona, Spain) following the manufacturer’s instructions. A negative control and a Mock Community control (Zymobiomics Microbial Community DNA) were included to ensure the quality of the analysis. The amplification of the samples was performed using specific primers for the V3–V4 regions of the 16S rRNA DNA (F5′-TCGTCGGCAGCGTCAGATGTGTATAAGAGACAGCCTACGGGNGGCWGCAG-3′, R5′GTCTCGTGGGCTCGGAGATGTGTATAAGAGACAGGACTACHVGGGTATCTAATCC-3′) [22]. The preparation of libraries was carried out in Microomics Systems SL (Barcelona, Spain). The amplification was performed after 25 PCR cycles using the Illumina Miseq sequencing 300 × 2 approach.

For sequencing data bioinformatics, the sequence reads generated were processed using QIIME version 2019.4 software [23]. The taxonomic assignment of phylotypes was performed using a Bayesian Classifier trained with Silva V4 database version 138 (99% OTU full-length sequences) [24]. A detailed description of all further steps in the bioinformatic analysis is available in our previous publication [25].

### 2.6. Statistical Analysis

All the statistical analyses were performed with open-source software R v4.0.3 (R Foundation for Statistical Computing, Vienna, Austria). Growth performance data were analyzed with ANOVA as a complete randomized design, together with Tukey’s test for multiple comparisons. Data of the SCFA concentrations in feces and cortisol and cortisone in hair were analyzed using a linear mixed model as a factorial arrangement (lactation growth x nursery growth). The model included lactation and nursery growth as fixed effects, their interaction, sex as a covariable, and the lactation mother as a random effect. Normality and homoscedasticity were checked with the Shapiro–Wilk test and Levene’s test, respectively. A Box–Cox transformation was performed on the cortisol concentration data, and a logarithmic transformation on cortisone and the cortisol-to-cortisone ratio for is statistical analysis after the Shapiro–Wilk test revealed it did not have a normal distribution. Alpha diversity was calculated using the vegan package [26] from raw counts (OTU level), including observed OTUs, Pielou’s Evenness, and the Shannon Index. An ANOVA test was performed to test group differences for the alpha diversity. A principal coordinates analysis (PCoA) was calculated using beta diversity distance matrices (Bray–Curtis). A permutational multivariate analysis of variance (PERMANOVA) was used to test the effects of day and treatment and their interaction on the Bray−Curtis distance between samples. Microbial diversity was analyzed as a factorial arrangement with treatment and sampling day as the main factors. A differential abundance analysis was performed using the metagenomeseq package [27] to examine differences in the genus level. Separate analyses were performed to analyze the effects of lactation and nursery growth. A cumulative sum scaling (CSS) [28] normalization of the raw counts was performed, and a zero-inflated Gaussian mixture model was used for the analysis. Relative abundances were used to plot taxon abundances at the phylum and family levels for each group. Log2 fold changes were calculated using relative abundances for the taxonomical groups that showed different abundances between the growth categories at each stage. *p*-Values for the differential abundance analysis were corrected by the false discovery rate (FDR). The pig was considered the statistical unit in all analyses, and statistical significance and tendencies were considered at *p* ≤ 0.05 and 0.05 < *p* ≤ 0.10, respectively. Significance and tendencies were considered using the FDR value instead of the *p*-value in the microbiota differential abundance analysis.

## 3. Results

The growth characteristics of the experimental groups and the results of the statistical comparisons are detailed in Table 1 (See Appendix A for additional information regarding the statistical analyses). Birth BW was significantly higher in the Fast_Fast group than in the Slow_Slow group (*p* < 0.05). In congruence with the experimental design, lactation growth and weaning BW were different between the fast lactation and slow lactation groups. Additionally, nursery growth and the final BW were different among all groups, starting with the highest values for the Fast_Fast group and followed by the Slow_Fast, Fast_Slow, and Slow_Slow groups.

### 3.1. Cortisol and Cortisone Concentrations in Hair

The cortisol and cortisone concentration levels in hair and the cortisol-to-cortisone ratio for each group are represented in Table 3. The cortisone levels were higher in the animals with a fast growth rate during nursery (*p* < 0.05) than their slowest counterparts. The cortisol-to-cortisone ratio also showed a tendency to be lower in the fast-growing pigs during the nursery period (*p* = 0.065). No differences between the sexes were identified (Appendix A).

The sample size for the cortisol analysis: Fast_Fast (19), Fast_Slow (19), Slow_Fast (19), and Slow_Slow (18). The sample size for cortisone and the cortisol/cortisone ratio analysis: Fast_Fast (*n* = 18, outliers = 1), Fast_Slow (*n* = 16, outliers = 3), Slow_Fast (*n* = 13, outliers = 6), and Slow_Slow (*n* = 13, outliers = 5). Outliers correspond to cortisone values below the limit of detection (<2 pg/mg).

### 3.2. Microbiota Structure and Biodiversity

After quality control, an average of 65,358 ± 14,527 high-quality reads for each fecal sample was generated. The microbial community alpha diversity measured as observed in the OTUs, Pielou’s Evenness Index, and Shannon Index are summarized in Table 4. The microbiota of the animals with fast growth during the nursery period showed a higher evenness (*p* < 0.05) than their slowest counterparts. No differences between the sexes were identified (Appendix A).

The sample size for the alpha diversity analysis: Fast_Fast (*n* = 19), Fast_Slow (*n* = 18, missing samples = 1), Slow_Fast (*n* = 17, missing samples = 2), and Slow_Slow (*n* = 17, missing samples = 1).

The beta diversity analysis at the OTU level using the Bray–Curtis distance revealed that the microbial structure of the animals was different according to their growth in both the lactation and nursery periods (*p* < 0.05). The principal coordinate analysis (PCoA) based on the Bray–Curtis distance matrix exposed a different clustering of the individuals according to their growth during each period, although the 2D representation only preserved 10.8% of the total variance (Figure 1). The sum of the first five principal coordinates preserved 21.72% of the variance.

### 3.3. Fecal Microbiota Composition

The microbiota composition of each group, represented as the relative abundance of the main phyla and families, is depicted in Figure 2. Firmicutes was the predominant phyla (80%), followed by Bacteroidota (11%), Patescibacteria (3%), and Actinobacteriota (3%), representing together 96% of the fecal microbiome. At the family level, Ruminococcaceae (14%), Erysipelotrichaceae (14%), the (Eubacterium) coprostanoligenes group (13%), and Lachnospiraceae (12%) were the predominant groups.

Pigs’ growth during lactation was associated with a few statistical differences in the main groups at the phylum and family levels. Microbial populations belonging to the Patescibacteria phyla showed higher abundance in the fast lactation growth group (3%) than the slow counterparts (2%; *p* < 0.05). Firmicutes was numerically lower in the pigs showing fast growth during the lactation period (Fast_Slow, 79%; Fast_Fast, 75%) compared to the ones showing slow growth (83%). This difference was mainly counterbalanced by an increased abundance of Bacteroidetes, as well as the abovementioned Patescibaceria phylum.

In Figure 3, the Log2 fold changes were calculated for the taxa that showed significant differences between the fast and slow growth groups in each period at the genus level. Fast growth during the lactation period was associated with a higher abundance of the genera *dgA-11* gut group, *Candidatus Saccharimonas*, *Colidextribacter*, *Bacteroidales RF16* group, *Campylobacter* (*p* < 0.05), and *Lachnospiraceae XPB1014* group (*p* = 0.054), while the fast growth during the nursery period was associated with a higher abundance of the genera *Streptococcus* (*p* < 0.05), *Dialister* (*p* = 0.056), *(Eubacetrium) ventriosum* group (*p* = 0.056), and *Lactobacillus* (*p* = 0.092). On the other hand, slow growth during the lactation period was associated with a higher abundance of the genus *(Eubacterium) ruminantium* group, *Lachnospiraceae UCG-002*, *Lachnospiraceae NK4A136* group (*p* < 0.05), *Candidatus Soleaferrea* (*p* = 0.054), and *Lachnospiraceae AC2044* group (*p* = 0.093), while the slow growth during nursery period was associated with the *Anaerofustis* (*p* < 0.05) and *Terrisporobacter* (*p* = 0.056) genera. All genera that showed differences between groups presented relative abundances < 1%, except for *Candidatus* Saccharimonas, which was 2.7%.

### 3.4. Short-Chain Fatty Acid Concentrations in Feces

The SCFA concentration in the fecal samples showed differences between groups, which are depicted in Figure 4. The analysis of SCFA in feces revealed a higher concentration of total SCFA (*p* < 0.05) and a tendency to have higher concentrations of acetate (*p* = 0.056) in animals experiencing fast growth during the lactation period. Furthermore, fast growth was associated with the highest levels of butyrate in both periods (*p* < 0.05). No differences were identified between groups in the analysis of propionate (15.8 ± SD 3.83 µmol/g), iso-butyrate (1.1 ± SD 0.54 µmol/g), iso-valerate (0.8 ± SD 0.38 µmol/g), valerate (1.4 ± SD 0.54 µmol/g), succinate (1.6 ± SD 0.39 µmol/g), and formate (5.1 ± SD 0.21 µmol/g) (*p* > 0.05), and no interactions were observed for any of the SCFA measured (*p* > 0.05). No differences between sexes were identified (Appendix A).

## 4. Discussion

In the present study, 75 pigs with different growth characteristics during early life were selected from a single weaning batch to determine their glucocorticoid levels in their hair, fecal microbiota composition, and fecal SCFA concentrations. The distribution of animals among the groups was performed according to the differences in growth during the lactation and nursery periods, while their BW was not considered. Due to that, there were differences in the birth BW among the experimental groups. Piglets within the Slow_Slow group were born smaller than the animals in the Fast_Fast group, which might be explained by the fact that piglets born small often remain stunted, presenting lower growth rates than their big counterparts [29,30]. Some light birth weight pigs have fewer muscle fibers [31]; they may show an insufficient colostrum intake [32], or they may have a retarded intestinal maturation [33], which might compromise their growth for the rest of their lives. Additionally, pigs in the Slow_Fast group were born numerically bigger than pigs in the Slow_Slow group. This is consistent with other studies that observed that piglets born with higher BW have a greater ability to show compensatory growth after having shown poor growth during the suckling phase [30,34]. In addition, the weaning BW influenced nursery growth. Animals with a high weaning BW had faster growth than their smallest counterparts during nursery, despite being selected for the same nursery growth group. It is necessary to consider these differences, because they might have an influence on the results of the present investigation.

### 4.1. Cortisol and Cortisone in Hair

As part of this study, the hair concentrations of cortisol and cortisone at the end of the nursery period were analyzed, and the ratio of cortisol to cortisone was calculated. The use of cortisol as a stress biomarker is an extended practice in research with pigs and humans [17]. The use of hair samples has some advantages over other biological samples, such as blood or saliva, in that it can be collected noninvasively, it can indicate long-term concentrations, and the manipulation of the subject at the time of collection does not interfere with the results [35,36]. The analysis revealed higher concentrations of cortisone than cortisol in the hair of the pigs, a fact that has also been reported in previous studies [19,36]. Cortisone and the cortisol-to-cortisone ratio in hair, which are considered estimators of the 11β-HSD type 2 activity, have attracted much attention as biomarkers of stress [19,35,36,37,38,39]. Although chronic stress is recognized to have negative effects on growth in early life [40], there are no previous exploratory studies aiming to establish a relationship between growth and the metabolism of glucocorticoids. Pigs showing a fast growth rate during the nursery period, especially animals in the Fast_Fast group, showed higher levels of cortisone and a tendency to have a lower cortisol-to-cortisone ratio, suggesting that they have a higher 11β-HSD type 2 activity than slow-growing pigs. The activity of 11β-HSD type 2 in hair, which converts active cortisol to inactive cortisone, has been considered a biomarker of chronic stress in different species. For instance, a previous study associated an increase in cortisone levels in the hair of sows with the stress experienced during the farrowing and lactation periods [35]. Additionally, research performed on humans associated psychosocial stress in young children with higher cortisone concentrations [37]. In addition, a study on sheep observed that a localized experimental infection could increase the deposition of cortisone in the hair at the site of infection [36]. According to these previous studies, pigs with the fastest growth may have experienced higher levels of stress, although their growth was not affected. A possible explanation for that result is that pigs with a higher growth rate also present a higher feed intake and are more exposed to the competition for feed. However, there are discrepancies in the literature regarding the association between the activity of 11β-HSD type 2 and chronic stress, especially in other sample types different than hair. Studies in humans have observed that elderly people with high cortisol-to-cortisone ratios in spot urine had higher levels of perceived stress [41] or that pregnant women with higher emotional support showed a higher metabolization of cortisol to cortisone in saliva [38]. Additionally, a decreased cortisol-to-cortisone ratio in 24-h urine was associated with an increase in performance in elite swimmers, which suggests that the increased inactivation of cortisol might protect the anabolic processes in the muscles against the deleterious effect of prolonged hypercortisolism [42]. Due to the discrepancies in the literature and the lack of complementary feeding and/or social behavioral information about the pigs in the study, we cannot reach any conclusion about the stress levels of the animals based on the glucocorticoid levels in their hair.

On the other hand, there is another isozyme participating in the metabolism of glucocorticoids, although it is discretely expressed in the hair follicle: 11β-HSD type 1 [43,44]. It catalyzes the reverse reaction to that performed by 11β-HSD type 2. Cortisol can be metabolized into cortisone by 11β-HSD type 2, and cortisone can be transformed back into cortisol by 11β-HSD type 1. In a previous study, we identified an increased expression of 11β-HSD type 1 in the intestines of piglets with lower BW both at birth and at weaning, which might indicate that fetal distress and the neonatal inflammatory condition can cause its long-term activation [45]. Therefore, although its activity is residual in hair follicles, we could speculate that 11β-HSD type 1 activity might be enhanced in slow-growing pigs, increasing the cortisol levels, the active form that exerts the inhibitory effects over muscle growth [46]. Additionally, 11β-HSD type 2 activity might be higher in fast-growing pigs, inactivating the cortisol to cortisone reaction. This hypothesis should be taken into closer consideration for future studies, as well as a possible gestational or neonatal imprinting of these two isoenzymes.

Differences in the birth BW between groups might have affected the glucocorticoid levels by possible gestational imprinting, but we cannot rule out a direct contribution of glucocorticoids deposited during the prenatal period, since the hair presented by the pigs at birth was not shaved due to the experimental logistics. Previous studies speculate that the time window for glucocorticoid deposition includes about one to two months prior to the date of collection, excluding the last 15 days [35], but the exact timing is uncertain, so it is not known whether the gestational period could influence the levels obtained. However, the contribution of the hair grown before birth to the total hair length at the time of collection is small, suggesting that glucocorticoids deposited during the gestational period might have a reduced impact on the total levels obtained at the end of the nursery period.

### 4.2. Fecal Microbiota Composition

The analysis of the fecal microbiota of the piglets in the study revealed that Firmicutes and Bacteroidetes were the dominant phyla, in agreement with other studies [47,48,49,50,51,52]. However, few studies have researched the possible link between intestinal microbiota at a particular age and the previous individual growth traits. The 16S rRNA gene sequence-based analysis of the feces of the piglets at the end of the nursery period revealed differences in the microbial diversity and composition between animals showing a previous fast and slow growth in each period. The beta diversity analysis showed differences in the microbial structure, in line with a previous study that observed differences in the microbiota of heavy and light pigs at d100 of life [53]. Additionally, pigs with a fast growth during the nursery period had a significantly higher evenness than their slowest counterparts, in agreement with Han et al. (2017), who reported higher microbial diversity in the microbiota of heavy 9-week-old pigs compared to lighter pigs [54]. Research in human children aged 0–2 years also revealed that a reduced microbial diversity was predictive of a future growth deficit [55]. However, this result conflicts with other studies that observed higher values in alpha diversity indexes in lighter pigs [53,56,57]. The relationship between intestinal microbial diversity and body weight is still a controversial issue.

#### 4.2.1. Lactation Growth Effect on Fecal Microbiota and SCFA Concentrations

The study of the relationship between lactation growth and intestinal microbiota composition at the end of the nursery period can help us to improve our understanding of the impact of early microbial colonization in the posterior phases. Both in humans and pigs, the gut microbiota structure undergoes drastic changes during the weaning phase, shifting from bacterial populations oriented to degrade metabolites present in the mother’s milk to bacterial populations adapted to degrade complex carbohydrates existent in the novel food offered at weaning [51,58]. Nevertheless, several studies suggest that events and interventions during the lactation period have the potential to induce long-lasting effects on the immune system programming, barrier function, and bacterial composition, which are preserved beyond weaning [1,25,59,60]. Studies in humans showed contrasting results regarding the long-term repercussions of the early modulation of the microbiome. However, interventional studies on animal models are more convincing in this respect [4]. For instance, in a previous report, we observed that the supplementation of xylooligosacharides to piglets exclusively during the lactation period stimulated the growth of fiber-degrading bacterial populations in the post-weaning period [25].

The current study revealed that piglets’ growth during the lactation period was associated with differences at the end of the nursery period in the fecal concentrations of SCFA and the relative abundance of certain microbial populations. First, although it was not a significant difference, piglets showing a fast growth during the lactation period had a numerically lower abundance of Firmicutes and higher abundance of Bacteroidetes than the slow-growth group. The ratio between Firmicutes and Bacteroidetes has been assumed to play a role in weight gain potential and the development of obesity [61]. Generally, the literature describes a higher Firmicutes/Bacteroidetes ratio in obese or overweight animals and humans [48,53,54,56,61,62,63]. Researchers explain these results by arguing that Firmicutes are more efficient in extracting energy from food than Bacteroidetes, thus promoting the more efficient absorption of calories and subsequent weight gain [62,63]. However, some reports did not observe any relationship or even a decreased Firmicutes/Bacteroidetes ratio in obese individuals [64,65]. On the other hand, McCormack et al. observed lower Firmicutes counts in weaned piglets, which showed better feed efficiency in the posterior phases [66], which could explain why this numerical difference was determined by lactation growth. Nevertheless, additional investigation is required to fully understand the relationship between the Firmicutes/Bacteroidetes ratio and the growth capacity of the host.

At the same time, piglets with a faster growth during lactation showed significantly higher levels of total SCFA in feces than the slow lactation growth group, independently of the nursery growth. The difference was mainly caused by the higher concentration of butyrate and the trend in the acetate levels in the fast-growth piglets during lactation. This result suggests that the microbiota of fast-growing pigs during lactation might have an increased capacity to produce SCFA from carbohydrate components present in the diet, which can be used as an extra energy source by the host. This is consistent with a study that reported a higher feed efficiency in pigs, with a higher total SCFA production in the caecum and a tendency to increase the butyrate ratio [67]. Furthermore, SCFAs have functional properties that might contribute to a superior growth capacity other than representing an extra energy supply. Butyrate plays an important role in cell proliferation and development, has regulatory functions on the metabolism, and is considered a health-promoting molecule. Additionally, it represents the preferred energy source for the colonocytes and helps to guarantee a correct gut function [68]. Additionally, acetate has an important regulatory role in body weight control and insulin sensitivity through its effects on lipid metabolism and glucose homeostasis [69]. However, during lactation, the main substrates for fermentation and SCFA production were the oligosaccharides in the sow’s milk, whereas, at the time of sampling, they were the complex carbohydrates present in the plant-based ingredients of the nursery diet. This might indicate that pigs showing a good growth during lactation developed a microbiota with a better capacity for fiber degradation and SCFA production in the posterior stages. Further research is necessary to confirm this long-term influence and its long-term impact on the digestive and immune function.

The relative abundance analysis of the bacterial genus associated with lactation growth revealed subtle changes in bacterial populations with abundances < 1%, with the only exception of *Candidatus Saccharimonas*, with a mean abundance of 2.7%, which had higher representativeness in fast-growing animals during lactation. This genus has a fermentative metabolism with acetate and lactate as the main products, which could have contributed to the higher concentration of acetate in the feces of these piglets.

#### 4.2.2. Nursery Growth Effect on Fecal Microbiota and SCFA Concentrations

Growth during the nursery period would be expected to have a greater impact on the microbiota at the end of this stage than lactation growth. Nevertheless, the analysis revealed fewer changes associated with the nursery growth. The abundance of Firmicutes and Bacteroidetes was not significantly different between the fast and the slow nursery growth groups, and the numerical difference was contemptible.

The total SCFA concentration was not affected by nursery growth. However, fast-growing pigs during nursery had significantly higher levels of butyrate in feces than their slower counterparts. An increased butyrate concentration was associated with fast growth during both the lactation and nursery periods, suggesting that it plays an important role in growth. Although it is difficult to identify the relevance of the causes and/or consequences associated with these changes on the gut microbiota and fermentation activity, the present findings may indicate that promoting the growth of butyrogenic bacterial populations might be a promising strategy for improving growth and redirect the trend of individuals with an impaired growth. This strategy has been tested in previous studies in swine. For instance, supplementing gestating sows with alfalfa meal, a highly fibrous ingredient, stimulated the growth of anti-inflammatory and butyrogenic bacteria in their gut, which led to an increase of butyrate concentration in their feces during lactation, as well as multiple benefits in the sows’ health and performance [70]. Alternatively, there is the possibility of supplementing exogenous butyrate for the organism. In fact, butyrate supplementation has already been explored in humans as a potential therapy for metabolic and intestinal diseases. In particular, due to its regulatory role of body weight gain and metabolism, it has been tested for treating obesity [71]. In pigs, the supplementation of sodium butyrate in the pigs’ diet proved to be a successful strategy to improve their growth performance, intestinal health, and morphology [72,73,74].

A relative abundance analysis of the bacterial genus based on nursery growth also showed fewer differences than for lactation growth. Nursery growth was associated with two significant differences versus the eight identified for lactation growth. These differences were identified in the microbial genera with low counts (<1% of relative abundance).

The aim of this study was to explore the characteristics of piglets with different adaptation capacities during the initial stages of life to identify opportunities to improve the growth of slow-growing individuals. This study only analyzed the differences among piglets showing extreme growth, which facilitated the detection of differences among groups with a limited number of animals sampled. However, this design did not provide information on the entire spectrum of pigs, which would have allowed the study of correlations among the different variables. In addition, sampling at various time periods throughout early life instead of only one time at the end of the nursery period would have provided a better understanding of the causal effects among the growth, gut microbiota, and stress during this period. Another limitation of the present study were the initial differences in birth BW among the growth groups. Birth BW is an important factor associated with the pre- and postnatal development of the pig, and these differences might have introduced a confounding factor that should be taken into consideration in the interpretation of the results. Moreover, the analysis of stress biomarkers was not accompanied by behavioral observations or other indicators of stress, which prevented us from drawing a conclusion about the stress levels of the studied pigs. Furthermore, although the impact of prenatal glucocorticoid deposition in the analyzed hair during the nursery period was expected to be low, the fact that the pigs were not shaved at the beginning of the experimental period implied some uncertainty in the time window represented in the results.

Despite the limitations already described, the obtained results indicated that the events that occurred during both the lactation and nursery periods influenced the glucocorticoid deposition in hair, gut microbiota, and its fermentation capacity. A glucocorticoid analysis revealed that fast-growing pigs might have an increased 11β-HSD type 2 activity, converting cortisol to cortisone, although its implications on the stress status need further confirmation. Regarding the microbiota and SCFA profiles, a larger number of differences were identified associated with lactation growth than nursery growth. This observation adds further evidence to the hypothesis that microbial colonization and modulation during the lactation period have a critical influence on the development and configuration of adult microbiota. In addition, the preservation of differences between fast and slow growers during lactation until the end of the nursery period might indicate that early interventions in the microbiota might have a long-term effect from weaning onwards. Most of the differences associated with growth were not consistent between the lactation and nursery periods, which calls into question the effectiveness of modulating the microbiota during early ages to obtain growth improvement during the posterior stages. However, the fecal butyrate levels were positively associated with growth during both periods, suggesting that stimulating butyrogenic microbiota during early life could provide an opportunity to redirect the growth of less efficient individuals. This study encourages the investigation of the implications of 11β-HSD type 2 activity on growth during early life, as well as the research of a possible gestational imprint on its activity. Additionally, future studies should examine the causal relationships among stress, microbiota, and growth, as well as effective strategies to modulate them to improve health and growth during early life.

## Figures and Tables

**Figure 1 nutrients-14-04639-f001:**
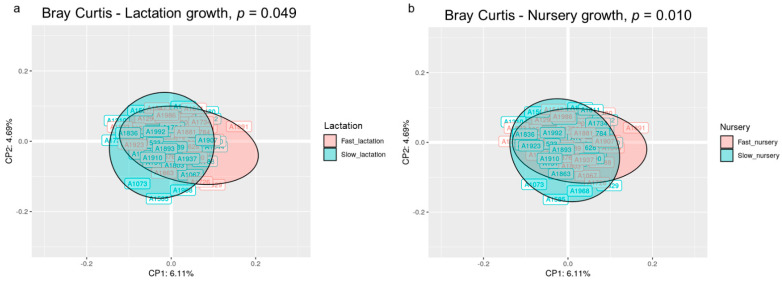
Principal coordinate analysis (PCoA) ordination performed by Bray–Curtis dissimilarities representing the microbiome compositions of all animals clustered by lactation growth (**a**) and nursery growth (**b**). The *p*-value corresponds to the permutational analysis of variance (PERMANOVA) test for each factor. Sample size for the PCoA: Fast_Fast (*n* = 19), Fast_Slow (*n* = 18, missing samples = 1), Slow_Fast (*n* = 17, missing samples = 2), and Slow_Slow (*n* = 17, missing samples = 1).

**Figure 2 nutrients-14-04639-f002:**
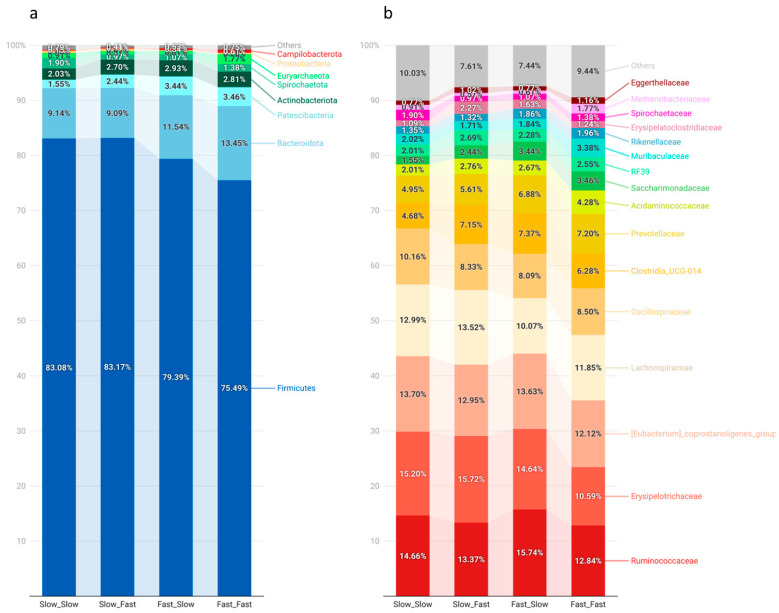
Relative abundances (RA) of the main phyla (**a**) and families (**b**) observed in the analysis of the microbiota of piglets by massive sequencing of the 16S rRNA gene. The figure was created with the online open-source tool Datawrapper (http://datawrapper.de; accessed on 2 November 2022). Fast_Fast: pigs showing fast growth during the lactation and nursery periods (*n* = 19), Fast_Slow: pigs showing fast growth during the lactation period and slow growth during the nursery period (*n* = 18, missing samples = 1), Slow_Fast: pigs showing slow growth during the lactation period and fast growth during the nursery period (*n* = 17, missing samples = 2), and Slow_Slow: pigs showing slow growth during the lactation and nursery periods (*n* = 17, missing samples = 1).

**Figure 3 nutrients-14-04639-f003:**
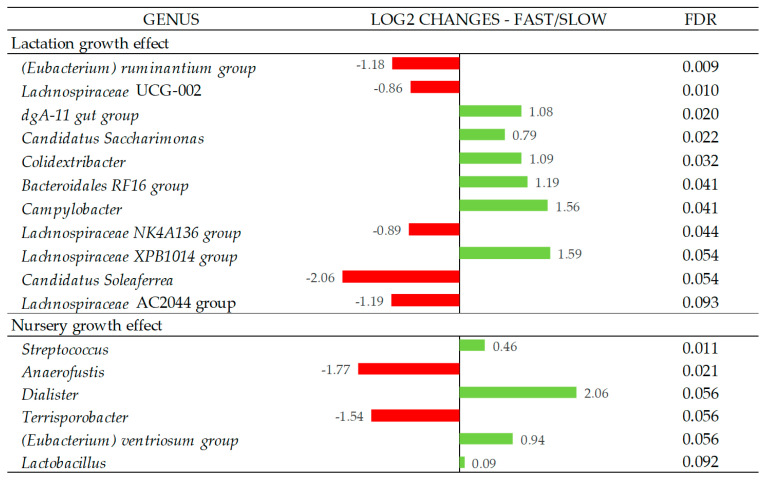
Log2 changes between the fast- and slow-growth pigs during the lactation and nursery periods (fold discovery rate *p*-adjusted < 0.1) in microbial genera. Taxa are sorted by the level of significance (from higher to lower). Differences presented are based on all taxa detected in the samples per group. Sample size for the differential abundance analysis: Fast_Fast (*n* = 19), Fast_Slow (*n* = 18, missing samples = 1), Slow_Fast (*n* = 17, missing samples = 2), and Slow_Slow (*n* = 17, missing samples = 1).

**Figure 4 nutrients-14-04639-f004:**
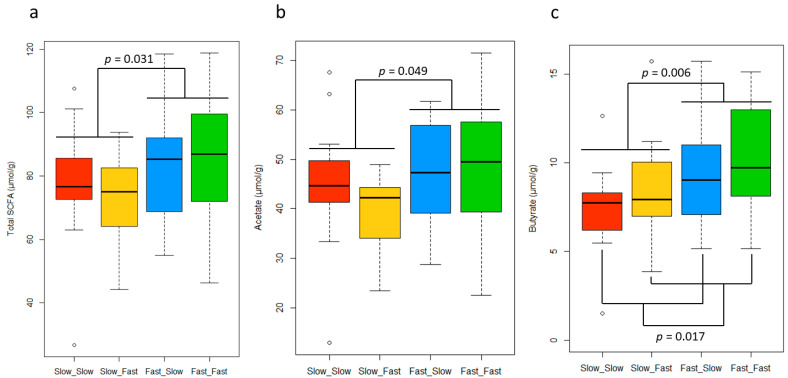
Boxplot representing the total short-chain fatty acid (SCFA) (**a**), acetate (**b**), and butyrate (**c**) concentrations in feces for each group. The *p*-values correspond to the factorial ANOVA. ast_Fast: pigs showing fast growth during the lactation and nursery periods (*n* = 18, missing samples = 1), Fast_Slow: pigs showing fast growth during the lactation period and slow growth during the nursery period (*n* = 18, missing samples = 1), Slow_Fast: pigs showing slow growth during the lactation period and fast growth during the nursery period (*n* = 16, missing samples = 3), and Slow_Slow: pigs showing slow growth during the lactation and nursery periods (*n* = 17, missing samples = 1).

**Table 1 nutrients-14-04639-t001:** Mean body weight (BW) and average daily gain (ADG) of the animals included in each group for the lactation and nursery periods. Each mean is followed by its corresponding SEM.

Group ^1^	Group Size (*n*)	Lactation ADG, g/d	Nursery ADG, g/d	Birth BW, kg	Weaning BW, kg	Nursery BW, kg
Fast_Fast	19	246 ± 5.2 ^a^	273 ± 7.1 ^a^	1.45 ± 0.026 ^a^	6.6 ± 0.09 ^a^	17.9 ± 0.33 ^a^
Fast_Slow	19	244 ± 4.6 ^a^	128 ± 3.7 ^c^	1.31 ± 0.069 ^ab^	6.4 ± 0.09 ^a^	11.7 ± 0.20 ^c^
Slow_Fast	19	148 ± 3.3 ^b^	238 ± 5.7 ^b^	1.35 ± 0.074 ^ab^	4.4 ± 0.11 ^b^	14.2 ± 0.31 ^b^
Slow_Slow	18	138 ± 5.9 ^b^	112 ± 3.5 ^d^	1.21 ± 0.056 ^b^	4.1 ± 0.11 ^b^	8.7 ± 0.17 ^d^
*p*-value		<0.001	<0.001	0.041	<0.001	<0.001

^1^ Fast_Fast: pigs showing fast growth during the lactation and nursery periods, Fast_Slow: pigs showing fast growth during the lactation period and slow growth during the nursery period, Slow_Fast: pigs showing slow growth during the lactation period and fast growth during the nursery period, and Slow_Slow: pigs showing slow growth during the lactation and nursery periods. ^a,b,c,d^ Values with different letters in the same column are significantly different, according to ANOVA and Tukey’s adjust.

**Table 2 nutrients-14-04639-t002:** Diets offered to the animals included in the trial.

Ingredient, %	Creep Feed	Pre-Starter	Starter
Wheat	17.8	23.0	17.5
Barley	7.0	10.0	30.7
Maize	1.2	1.4	21.7
Rapeseed	-	-	3.0
Sweet milk whey	15.0	10.0	-
Broken rice	15.0	15.0	-
Spray dried yogurt	13.2	-	-
Soy protein concentrate	8.4	3.7	-
Soybean meal 47% crude protein (CP)	-	13.0	-
Soybean meal 44% CP	-	-	12.3
Extruded soybeans	-	6.0	-
Porcine digestible peptides 62% CP	5.8	2.1	-
Porcine digestible peptides 50% CP	-	-	5.7
Animal plasma 80% CP	1.7	1.7	-
Skimmed milk	3.3	-	
Extruded cereals ^1^	3.0	-	-
Lard	1.0	1.5	3.3
Lactose	-	2.8	-
Sucrose	2.2	1.7	-
Sugar beet pulp	2	1.8	0.5
Wheat bran	1.6	3.8	-
Lignocellulose 65% crude fiber (CF)	-	-	1
Vit-Min premix ^2^	0.5	0.5	0.5
Liquid lysine 50%	0.61	0.62	0.99
DL-Methionine	0.26	0.24	0.22
L-Threonine	0.19	0.18	0.27
L-Valine	0.16	0.08	0.09
L-Tryptophane	0.10	0.07	0.07
Histidine	-	-	0.11
Isoleucine	-	-	0.03
Mono calcium phosphate	-	0.42	1.10
Calcium carbonate	-	0.19	0.71
Salt	-	0.26	0.25
Calculated composition			
NE, kcal/kg	2558	2430	2500
Ash, %	5.1	5.0	5.2
Crude protein, %	19.7	18.9	17.5
Ether extract, %	4.7	4.1	6.6
Crude fiber, %	1.9	3.0	5.4
Starch, %	29.8	30.0	39.1
Calcium, %	0.550	0.650	0.700
Total *p*, %	0.484	0.515	0.626
Digestible *p*, %	0.333	0.300	0.423
Digestible amino acids			
Lys, %	1.372	1.294	1.187
Met, %	0.578	0.497	0.453
Met + Cys, %	0.822	0.774	0.712
Thr, %	0.891	0.839	0.794
Trp, %	0.301	0.284	0.251

^1^ Composition: 50% maize, 30% barley, and 20% wheat. ^2^ Provided per kilogram of diet: 12,000 IU of vitamin A (acetate), 2000 IU of vitamin D3 (cholecalciferol), 250 IU of vitamin D (25-hydroxicholecalciferol), 75 mg of vitamin E, 2 mg of vitamin K3, 3 mg of vitamin B1, 7 mg of vitamin B2, 7.33 mg of vitamin B6, 15 mg of vitamin B12, 17 mg of D-pantothenic acid, 45 mg of niacin, 0.2 mg of biotin, 1.5 mg of folacin, 80 mg of Fe (chelate of amino acids), 100 mg of Zn (chloride), 12.5 mg Zn (chelate of amino acids), 12.5 mg of Mn (chloride), 0.3 mg of Se (inorganic), 2.04 mg of BHT, 400 UI of endo-1,4 beta-xylanase, and 250 OTU of 6-phytase.

**Table 3 nutrients-14-04639-t003:** Cortisol and cortisone concentration levels in hair and calculated cortisol-to-cortisone ratio for each growth group.

Item		Cortisol (pg/mg)	Cortisone (pg/mg)	Cortisol/Cortisone
Lactation Growth	Nursery Growth			
Fast	Fast	19.2	188	0.27
	Slow	17.4	86	0.54
Slow	Fast	20.3	105	0.56
	Slow	19.1	115	0.67
SEM ^1^		1.06	33.1	0.163
Lactation growth				
Fast		18.3	140	0.40
Slow		19.7	110	0.62
SEM		0.74	22.3	0.115
Nursery growth				
Fast		19.7	153	0.39
Slow		18.2	99	0.60
SEM		0.74	23.4	0.110
*p*-value ^2^				
Lactation growth		0.130	0.329	0.227
Nursery growth		0.463	0.041	0.069
Lactation × Nursery growth		0.852	0.174	0.157

^1^ Standard error of the mean. ^2^
*p*-value obtained from the factorial ANOVA test.

**Table 4 nutrients-14-04639-t004:** Alpha diversity measured as observed in the OTUs, Pielou’s Evenness Index, and Shannon Index for pigs showing fast and slow growth during the lactation and nursery periods.

Item	Observed OTUs	Evenness	Shannon
Lactation growth			
Fast	354	0.786	6.63
Slow	343	0.781	6.55
SEM ^1^	14.9	0.0078	0.109
Nursery growth			
Fast	353	0.797	6.72
Slow	345	0.770	6.46
SEM	15.1	0.0074	0.105
*p*-value ^2^			
Lactation growth	0.330	0.568	0.384
Nursery growth	0.969	0.016	0.127
Lactation × Nursery growth	0.526	0.985	0.904

^1^ Standard error of the mean. ^2^
*p*-values obtained from the factorial ANOVA test.

## Data Availability

The raw sequencing data employed in this article was submitted to the NCBI’s sequence read archive (https://www.ncbi.nlm.nih.gov/sra; accessed on 2 November 2022) and BioProject: PRJNA881848.

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
