# Peer review of "Fecal Microbiota and Hair Glucocorticoid Concentration Show Associations with Growth during Early Life in a Pig Model"

_nutrients, 2022, doi:10.3390/nu14214639_

Round 1
Reviewer 1 Report
The study is very interesting, and there are some comments:
1. For ADG, gender is an important factor. Whether it is considered in the analysis and how many boars and sows are in each group should be explained.
2. The performance of piglets during lactation period has a strong maternal effect. Is it considered? How to eliminate the influence of maternal effect?
3. In line 87, weaning day is d21, but in line 122, "the weaning(d28), piglets were moved......", Why need the interval of one week?
4. In my oppion, when the piglets are transferred to the nursery, there will be a problem of rebuilding the group order. The impact on different individuals will vary greatly, especially the ones that grow fast. How to evaluate?
Reviewer 2 Report
The manuscript "Faecal microbiota and hair glucocorticoid concentration show associations with growth during early life in a pig model" offers helpful information about pig performance, physiological stress response, and microbiota dynamics during the lactation and nursery period. The findings reveal that both periods influence glucocorticoid levels in hair, intestinal microbiota, and fermentation capacity, which is highly relevant for animal health and welfare. In addition, assessing early pig performance, health, and welfare in a reliable and robust manner will serve us in studying human development< therefore, in both ways, the manuscript goes ahead in a further step.
The authors know the area and have plenty of experience in animal studies. They wrote the manuscript very well. The manuscript is original, objective, and current and covers an important field of research.
My review is in the attached file. I encourage the authors to include information about the methods' validation, and the potential relationships between variables. I am also curious why you did not include the sex factor in the study. Finally, I invite the authors to acknowledge the study's limitations and the next steps in that research direction.
Best regards,
